# PUUPL: Positive-Unlabeled Learning with Uncertainty-aware Pseudo-label Selection

## Abstract

Positive-unlabeled (PU) learning aims at learning a binary classifier from only positive and unlabeled training data. Recent approaches addressed this problem via cost-sensitive learning by developing unbiased loss functions, and their performance was later improved by iterative pseudo-labeling solutions. However, such two-step procedures are vulnerable to incorrectly estimated pseudo-labels, as errors are propagated in later iterations when a new model is trained on erroneous predictions. To mitigate this issue we propose *PUUPL*, a new loss-agnostic training procedure for PU learning that incorporates epistemic uncertainty in pseudo-labeling. Using an ensemble of neural networks and assigning pseudo-labels based on high confidence predictions improves the reliability of pseudo-labels, increasing the predictive performance of our method and leading to new state-of-the-art results in PU learning. With extensive experiments, we show the effectiveness of our method over different datasets, modalities, and learning tasks, as well as improved robustness over misspecifications of hyperparameters and biased positive data. The source code of the method and all the experiments are available in the supplementary material.

## 1 Introduction

Many real-world applications involve positive and unlabeled (PU) datasets in which only some of the data is labeled positive while the majority is unlabeled and contains both positives and negatives. PU learning aims to learn a binary classifier in this challenging setting without any labeled negative examples. Learning from PU data can reduce deployment costs in many deep learning applications that otherwise require annotations from experts such as medical image diagnosis (Armenian and Lilienfeld, 1974) and protein function prediction (Gligorijević et al., 2021), and it can even enable applications in settings where the measurement technology itself can not detect negative examples (Purcell et al., 2019).

Some recent approaches such as unbiased PU (Du Plessis et al., 2014, uPU) and non-negative PU (Kiryo et al., 2017, nnPU) formulate this problem as cost-sensitive learning. Others approach PU learning as a two-step procedure first identifying and labeling some reliable negative examples, and then re-training the model based on this newly constructed labeled dataset (Bekker and Davis, 2020). These approaches show similarities with pseudo-labeling in semi-supervised classification settings (Lee, 2013).

Such pseudo-labeling techniques are however especially vulnerable to incorrectly assigned labels of the selected examples as these errors will propagate and magnify in the retrained model, resulting in a negative feedback loop. Worse yet, since the true labels are unavailable in PU learning, this situation is hard to detect by any metrics computed on the training set. This erroneous selection of unreliable pseudo-labels occurs when wrong model predictions are associated with excessive model confidence. Such poor model calibration is accompanied by a distortion of the signal for the pseudo label selection (Van Engelen and Hoos, 2020).

In recent literature on pseudo-labeling, this problem is recognized and successfully addressed by explicitly estimating the prediction uncertainty (Abdar et al., 2021; Rizve et al., 2021; Arazo et al., 2020). While this is the case for semi-supervised classification, there does not yet exist a method that explores the use of uncertainty quantification for pseudo-labeling in a PU learning context.

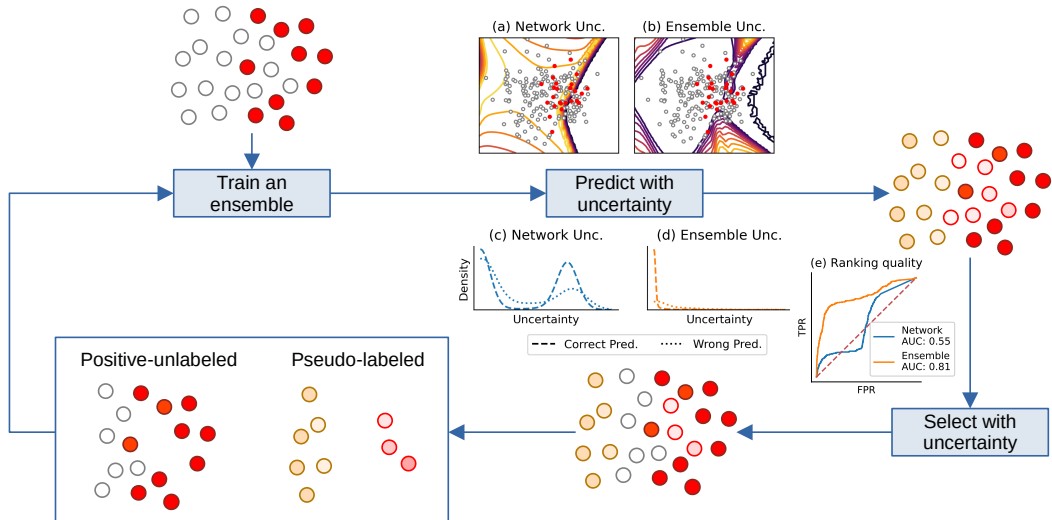

Figure 1: *PUUPL* is a pseudo-labeling framework for PU learning that uses the epistemic uncertainty of an ensemble to select confident examples to pseudo-label. The ensemble can be trained with any PU loss for PU data while minimizing the cross-entropy loss on the previously assigned pseudo-labels. On a toy example, a single network is not very confident on most of the unlabeled data (a), resulting in many high-confidence incorrect predictions and many low-confidence correct ones (c). The epistemic uncertainty of an ensemble is, on the other hand, very low on most of the the unlabeled data (b), resulting in most correct predictions having low uncertainty and most incorrect predictions having high uncertainty (d). Thus, the uncertainty of an ensemble can be used more reliably to rank predictions and select correct ones (e).

**Contributions:** Motivated by this, we propose a novel, uncertainty-aware pseudo-labeling framework for PU learning that uses established uncertainty quantification techniques to identify reliable examples to pseudo-label (Fig. 1). In particular, our contributions are: (1) We introduce *PUUPL* (Positive-Unlabeled, Uncertainty-Aware Pseudo-Labeling), a simple uncertainty-aware pseudo-labeling framework for PU learning. (2) *PUUPL* can use any loss function for PU learning, improving model performance while being robust to the specific data biases that the respective loss considers. (3) We evaluate our methods on a wide range of benchmarks and PU datasets, achieving state-of-the-art performance in PU learning. (4) Our extensive ablation studies provide new insights into uncertainty-aware pseudo-labeling for PU learning. Further, they show that our method is robust to the choices of hyperparameters, with 1% or less variability in test accuracy among different choices as well as distribution shifts between labeled positives in the train and test datasets. To the best of our knowledge, PUUPL is the first framework for PU learning which leverages uncertainty information during pseudo-labeling.

## 2 RELATED WORK

**PU Learning** PUL was introduced as a variant of binary classification (Liu et al., 2003) and is related to one-class learning (Ruff et al., 2018; Li et al., 2010), multi-positive learning (Xu et al., 2017), multi-task learning (Kaji et al., 2018), and semi-supervised learning (Chapelle et al., 2009). There exist three main research branches for PUL: two-step techniques, class prior incorporation, and biased PUL (Bekker and Davis, 2020). In this work, we combine Pseudo-Labeling which has similarities to two-step techniques, with biased PUL, also coined as reweighting methods, and refer to Bekker and Davis (2020) for a comprehensive overview of the field. In this context, Du Plessis et al. (2014) introduced the unbiased risk estimator uPU. Kiryo et al. (2017) showed this loss function is prone to overfitting in deep learning contexts as it lacks a lower bound and proposed the non-negative risk estimator nnPU as a remedy. Follow-up work on loss functions for PUL has focused on robustness w.r.t biases in the sampling process such as PUSB (Kato et al., 2019), PUbN (Hsieh et al., 2019) or PULNS (Luo et al., 2021).

**Uncertainty-aware Pseudo-Labeling** Pseudo-labeling follows the rationale that the model leverages its own predictions on unlabeled data as pseudo training targets to enable iterative semi-supervised model training. The first such approach for deep learning was introduced by Lee (2013), simply selecting the class with the highest predicted probability as a pseudo label. One weakness of pseudo-labeling is that erroneously selected pseudo-labels can amplify for training, potentially leading to model degradation. This is grounded in poor model calibration distorting the signal for the pseudo label selection (Van Engelen and Hoos, 2020). Iscen et al. (2019) try to mitigate this issue using confidence and class weights. Shi et al. (2018) use confident scores based on the geometric neighborhood of the unlabeled samples while Arazo et al. (2020) effectively tackle this *confirmation bias* using Mixup (Zhang et al., 2017), Label Noise (Tanaka et al., 2018), and Entropy Regularization (Grandvalet et al., 2005). Rizve et al. (2021) introduced a pseudo-labeling framework using a weighting scheme for class balancing and MC dropout (Gal and Ghahramani, 2016) for calibration, while Beluch et al. (2018) found deep ensembles (Lakshminarayanan et al., 2017a) to yield the best model calibration in an active learning context, especially in low-label regimes. The commonality of these works is the explicit consideration of model uncertainty to improve pseudo-label selection, which motivates us to apply this in the context of PU learning.

**Pseudo-Labeling for PU Learning** Two-step approaches in PU learning first identify negative samples from the unlabeled dataset, and then train a binary classification model on the original dataset augmented with the newly identified negatives (Bekker and Davis, 2020). These approaches share similarities with pseudo-labeling but lack an iterative feedback loop after the completion of the second step.

A first attempt to combine pseudo-labeling with PU learning was made with Self-PU (Chen et al., 2020b), where self-paced learning, a confidence-weighting scheme based on the model predictions and a teacher-student distillation approach are combined. Via this complex training scheme, Self-PU was shown to marginally outperform recent baselines. With *PUUPL*, we propose an alternative PL strategy for PU learning that performs better in a simpler and more principled way using implicitly well-calibrated models to improve the pseudo-label selection.

**Uncertainty-aware pseudo-labeling for PU learning** To the best of our knowledge, we are the first to introduce an uncertainty-aware pseudo-labeling paradigm to PU learning. Although our method shares the same motivation as that from Rizve et al. (2021) for semi-supervised classification, we differ in several important aspects: (1) we specifically target PU data with a PU loss, (2) we quantify uncertainty with an ensemble instead of Monte Carlo dropout, (3) we use epistemic uncertainty instead of the predicted class probabilities for the selection, (4) we do not use temperature scaling, and (5) we use soft labels.

## 3 METHOD

We propose *PUUPL* (Positive Unlabeled, Uncertainty-aware Pseudo-Labeling), an iterative pseudo-labeling procedure to progressively select and label the most confident examples from unlabeled data. The pseudo-code for *PUUPL* is shown as Algorithm 1. Our method separates the training set $X^{tr}$ into the sets $P$, $U$, and $L$ which contain the initial positives, the currently unlabeled, and the pseudo-labeled samples respectively. The set $L$ is initially empty. At each pseudo-labeling iteration, we first train our model using all samples in $P$, $U$, and $L$ until some convergence condition is met (Section 3.2). Then, samples in $U$ are predicted and ranked w.r.t their predictive uncertainty (Section 3.3) and samples with the most confident score are assigned the predicted label and moved into the set $L$ (Section 3.4). Similarly, samples in $L$ are also predicted and the most uncertain samples are moved back to the unlabeled set $U$ (Section 3.5). Next, the model is re-initialized to the same initial weights and a new pseudo-labeling iteration starts.

In the following, we first describe the notation used in this paper and then explain in detail the training procedure of *PUUPL*.

### 3.1 NOTATION

Consider input samples $X$ with label $y$ and superscripts $\cdot^{tr}$, $\cdot^{va}$ and $\cdot^{te}$ for training, validation, and test data respectively. The initial training labels $y^{tr}$ are set to one for all samples in $P$ and zero for all others in $U$. We group the indices of original positives, unlabeled, and pseudo-labeled samples

---

**Algorithm 1** Pseudocode for the PUUPL Training Procedure

---

**Input**

- Train, validation and test data $X^{tr}, y^{tr}, X^{va}, y^{va}, X^{te}, y^{te}$

- Number $K$ of networks in the ensemble (suggested $K = 2$)

- Maximum number $T$ of pseudo-labels to assign at each round (suggested $T = 1000$)

- Maximum uncertainty threshold $t_l$ to assign pseudo-labels (suggested $t_l = 0.05$)

- Minimum uncertainty threshold $t_u$ to remove pseudo-labels (suggested $t_u = 0.35$)

**Output** Model parameters $\theta^*$
1:  $P \leftarrow$ indices of positive samples in $X^{tr}$
2:  $U \leftarrow$ indices of unlabeled samples in $X^{tr}$
3:  $L \leftarrow \emptyset$                                               $\triangleright$ Indices of pseudo-labeled samples
4:  $\theta^0 \leftarrow$ Random weight initialization
5: **while** not converged **do**
6:     Initialize model weights to $\theta^0$                              $\triangleright$ Training
7:     Train an ensemble of $K$ networks on $X^{tr}, y^{tr}$ using the loss in Eq. 1
8:     Validate on $X^{va}, y^{va}$ and update weights $\theta^*$ if accuracy improved
9:     $\hat{f} \leftarrow$ ensemble predictions for $X^{tr}$                      $\triangleright$ Uncertainty
10:    Compute epistemic uncertainty $\hat{u}^e$ with $\hat{f}$ via Eq. 6
11:    $L^{\text{new}} \leftarrow$ Balanced set of examples to pseudo-label via Eq. 7 using $\hat{u}^e_U$, $T$ and $t_l$   $\triangleright$ Pseudo-labeling
12:    $U^{\text{new}} \leftarrow$ Examples to pseudo-unlabel via Eq. 10 using $\hat{u}^e_L$ and $t_u$
13:    $L \leftarrow L \cup L_b^{\text{new}} \setminus U^{\text{new}}$                           $\triangleright$ Update indices
14:    $U \leftarrow U \setminus L_b^{\text{new}} \cup U^{\text{new}}$
15:    $y_{L^{\text{new}}} \leftarrow \hat{p}_{L^{\text{new}}}$                                $\triangleright$ Update pseudo-labels
16:    $y_{U^{\text{new}}} \leftarrow 0$
17: **end while**
18: Restore the weights $\theta^*$ that scored highest on the validation set
19: Compute accuracy on the held-out test set $X^{te}, y^{te}$
20: **return** $\theta^*$

---

in $X^{tr}$ into the sets $P$, $U$, and $L$ respectively. Our proposed model is an ensemble of $K$ deep neural networks whose random initial weights are collectively denoted as $\theta^0$. The predictions of the $k$-th network for sample $i$ are indicated with $\hat{p}_{ik} = \sigma(\hat{f}_{ik})$, with $\sigma(\cdot)$ the logistic function and $\hat{f}_{ik}$ the predicted logits. The logits and predictions for a sample averaged across the networks in the ensemble are denoted by $\hat{f}_i$ and $\hat{p}_i$ respectively. We subscript data and predictions with $i$ to index individual samples, and use an index set in the subscript to index all samples in the set (e.g., $X^{tr}_U = \{X^{tr}_i | i \in U\}$ denotes the features of all unlabeled samples). We denote the total, epistemic and aleatoric uncertainty of sample $i$ as $\hat{u}^t_i$, $\hat{u}^e_i$, and $\hat{u}^a_i$, respectively.

## 3.2 Loss function

We train our proposed model with a loss function $\mathcal{L}$ that is a convex combination of a loss $\mathcal{L}_{PU}$ for the samples in the positive and unlabeled set ($P \cup U$) and a loss $\mathcal{L}_L$ for the samples in the pseudo-labeled set ($L$):

$$\mathcal{L} = \lambda \cdot \mathcal{L}_L + (1 - \lambda) \cdot \mathcal{L}_{PU} \tag{1}$$

with $\lambda \in (0, 1)$. The loss $\mathcal{L}_L$ is the binary cross-entropy computed w.r.t the assigned pseudo-labels $y$. Our model is agnostic to the specific PU loss $\mathcal{L}_{PU}$ used. This allows our method to be easily adapted to different scenarios for which a PU loss was proposed and improve over its performance, for example when coping with a selection bias in the positive examples (Kato et al., 2019) or the availability of a biased negative set (Hsieh et al., 2019). For the standard setting of PU learning, we use the non-negative correction $nnPU$ of the PU loss (Kiryo et al., 2017):

$$\mathcal{L}_{PU} = \pi \cdot \ell(P, 1) + \max\{0, \ell(U, -1) - \pi \cdot \ell(P, -1)\} \tag{2}$$

With $\pi$ the prior probability of a sample being positive, which we assume known and can be estimated from PU data (du Plessis et al., 2016), and $\ell(S, y)$ the expected sigmoid loss of samples in the set $S$ with label $y$:

$$\ell(S, y) = \frac{1}{|S|} \sum_{i \in S} \frac{1}{1 + \exp(y \cdot \hat{p}_i)} \tag{3}$$

### 3.3 MODEL UNCERTAINTY

To quantify the predictive uncertainty, we utilize a deep ensemble with $K$ networks with the same architecture, each trained on the full training dataset (Lakshminarayanan et al., 2017b). Given the predictions $\hat{p}_{i1}, \ldots, \hat{p}_{iK}$ for a sample $x_i$, we associate three types of uncertainties to $x_i$'s predictions (Hüllermeier and Waegeman, 2021): the aleatoric uncertainty as the mean of the entropy of the predictions (Eq. 4), the total uncertainty as the entropy of the mean prediction (Eq. 5), and the epistemic uncertainty formulated as the difference between the two (Eq. 6).

$$\hat{u}_i^a = -\frac{1}{K} \sum_{k=1}^{K} [\hat{p}_{ik} \log \hat{p}_{ik} + (1 - \hat{p}_{ik}) \log(1 - \hat{p}_{ik})] \tag{4}$$

$$\hat{u}_i^t = -\hat{p}_i \log \hat{p}_i - (1 - \hat{p}_i) \log(1 - \hat{p}_i) \tag{5}$$

$$\hat{u}_i^e = \hat{u}_i^t - \hat{u}_i^a \tag{6}$$

Epistemic uncertainty corresponds to the mutual information between the parameters of the model and the true label of the sample. Low epistemic uncertainty thus means that that the model parameters would not change significantly if trained on the true label, suggesting that the prediction is indeed correct. Using such a prediction as target in the cross entropy loss would in turn provide a stronger, more explicit learning signal to the model, so that a correctly pseudo-labeled example provides a larger decrease in risk compared to using the same example without any label within the positive-unlabeled loss.

### 3.4 PSEUDO-LABELING

The estimated epistemic uncertainty (Eq. 6) is used to rank samples of the unlabeled set $U$ and to select reliable samples for pseudo-labeling. Next, the predictions of the unlabeled samples $U$ are ranked according to their epistemic uncertainty (Eq. 6). Let $\rho(i)$ denote the rank of sample $i$, then the set $L^{\text{new}}$ of newly pseudo-labeled samples is formed by taking the $T$ samples with lowest uncertainty from $U$, ensuring that it is lower than the threshold $t_l$:

$$L^{\text{new}} = \{i \in U | \rho(i) \le T \wedge u_i^e \le t_l\} \tag{7}$$

Previous works have shown that balancing the pseudo-label selection between the two classes, i.e., ensuring that the ratio of newly labeled positives and negatives is close to a given target ratio $r$, is beneficial (Rizve et al., 2021). In this case, the set $L^{\text{new}}$ should be partitioned according to the model's predictions into a set $L_+^{\text{new}}$ of predicted positives and $L_-^{\text{new}}$ of predicted negatives, and the most uncertain samples in the larger set should be discarded to reach the desired ratio $r$, which we fix to 1. We then assign soft pseudo-labels, i.e., the average prediction in the open interval $(0, 1)$, to these samples:

$$y_i = \hat{p}_i \quad \forall i \in L_-^{\text{new}} \cup L_+^{\text{new}} \tag{8}$$

### 3.5 PSEUDO-UNLABELING

Similar to the way that low uncertainty on an unlabeled example indicates that the prediction can be trusted, high uncertainty on a pseudo-labeled example indicates that the assigned pseudo-label might not be correct after all. To avoid training on such possibly incorrect pseudo-labels, we move the pseudo-labeled examples with uncertainty above a threshold $t_u$ back into the unlabeled set:

$$U^{\text{new}} = \{i \in L | \hat{u}_i^e \ge t_u\} \tag{9}$$

$$y_i = 0 \quad \forall i \in U^{\text{new}} \tag{10}$$

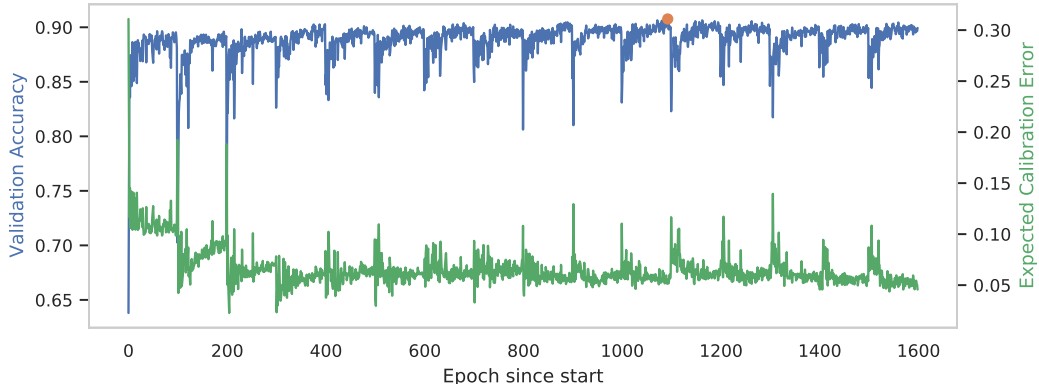

Figure 2: Validation accuracy (left, blue) and expected calibration error (ECE, right, green) for a run on CIFAR-10. The ensemble is trained for a fixed number of 100 epochs before pseudo-labeling, visible as the periodic spikes in both curves. Note the substantial reduction in ECE in the second and third pseudo-labeling iterations, when the ensemble is trained on soft labels. The best validation accuracy of 90.76% is indicated by the orange dot and corresponds to a test accuracy of 90.35%.

## 4 EXPERIMENTS

### 4.1 EXPERIMENTAL PROTOCOL

To empirically compare our proposed framework to existing state-of-the-art losses and models, we followed standard protocols for PU learning (Chen et al., 2020b; Kiryo et al., 2017; Kato et al., 2019).

**Datasets:** We evaluated our method in the standard setting of MNIST (Deng, 2012) and CIFAR-10 (Krizhevsky et al., 2009) datasets, as well as Fashion MNIST (F-MNIST) (Xiao et al., 2017), STL-10 (Coates et al., 2011) and IMDb (Maas et al., 2011) datasets to show the applicability to different data modalities. Similar to previous studies (Chen et al., 2020b; Kiryo et al., 2017; Kato et al., 2019), positives were defined as odd digits in MNIST, vehicles in CIFAR-10 and STL-10, and we used trousers, dress, sandals, sneaker, and ankle boots for F-MNIST and positive reviews for IMDb. For STL-10 we used all available labeled and unlabeled data and the official ten cross-validation folds. For all other datasets, we reserved a validation set of 5,000 samples and use all other samples for training with 1,000 randomly chosen labeled positives, as is common practice, and evaluated on the canonical test set of each dataset. More details are provided in Appendix B

**Network architectures:** To ensure a fair comparison with other works in PU learning (Chen et al., 2020b; Kiryo et al., 2017) we used the same architectures on the same datasets, namely a 13-layers convolutional neural network for the experiments on CIFAR-10 (Table A.3) and a MLP with four fully connected hidden layers of 300 neurons each and ReLU activation for MNIST and F-MNIST. For IMDb we used a bidirectional LSTM network with a MLP head whose architecture was optimized as part of the hyperparameter search.

**Training:** We trained all models with the Adam optimizer (Kingma and Ba, 2015) with $\beta_1 = 0.9$ and $\beta_2 = 0.999$, and an exponential learning rate decay with $\gamma = 0.99$. We further used the nnPU loss (Kiryo et al., 2017) as $\mathcal{L}_{PU}$ (Eq. 1) unless otherwise stated. As is common in the pseudo-labeling literature (Chen et al., 2020b; Rizve et al., 2021; Kato et al., 2019; Tanaka et al., 2018; Hu et al., 2021), we assume that a positive and negative labeled validation set is available and use this validation set for early stopping, i.e., stop the pseudo-labeling loop when the model's accuracy on this set has stopped improving, and use the parameters that achieved the highest validation accuracy to compute the test performance. An experiment will show how this requirement can be relaxed in practice.

**Hyperparameter tuning:** We used the Hyperband algorithm (Li et al., 2017) to optimize all hyperparameters on the CIFAR-10 dataset with $\eta = 3$ and $S = 4$, using the validation accuracy as the criterion to be optimized. The configuration that achieved the highest validation accuracy was

|          | MNIST        | F-MNIST      | CIFAR-10 (1,000 Lab.) | CIFAR-10 (3,000 Lab.) | IMDb         | STL-10       |
|----------|--------------|--------------|-----------------------|-----------------------|--------------|--------------|
| nnPU*    | 93.41 (0.20) | -            | 88.60 (0.40)          | -                     | -            | -            |
| SelfPU*  | 94.21 (0.54) | -            | 89.68 (0.22)          | 90.77 (0.21)          | -            | -            |
| DAN*     | -            | -            | -                     | 89.70                 | -            | -            |
| PAN*     | -            | -            | 89.70                 | -                     | 78.84        | -            |
| nnPU     | 95.26 (0.55) | 95.70 (0.18) | 89.20 (0.29)          | 90.91 (0.24)          | 78.57 (0.68) | 93.09 (0.54) |
| +PL      | 95.78 (0.30) | 95.84 (0.16) | 89.74 (0.46)          | 91.01 (0.18)          | 80.64 (0.99) | -            |
| +*PUUPL* | **96.01 (0.29)** | **95.91 (0.20)** | **90.18 (0.15)** | **91.44 (0.29)** | **81.83 (0.24)** | **93.30 (0.42)** |

Table 1: Test accuracy and standard deviation over five (ten for STL-10) runs in brackets on various datasets of recent methods for PU learning. The row "+PL" refers to an uncertainty-unaware pseudo-labeling baseline that uses the average prediction of an ensemble of networks as selection criterion, while the bottom row refers to our uncertainty-aware solution. We used 1,000 labeled positives for training except for CIFAR-10, where we also report the performance with 3,000 labeled positives, and STL-10, where we used the official ten cross-validation folds. Bold font indicates highest performance. *scores as reported by Chen et al. (2020b) and Hu et al. (2021) due to unavailability of source code.

then used as a basis for the ablation studies and fine-tuned on the remaining datasets to show that the pseudo-labeling hyperparameters (Table A.1) do not require tuning when transferred to other datasets and data modalities. Specifically, on the other datasets we only tuned hyperparameters related to network training such batch size, learning rate, weight decay, number of training epochs, dropout probability and other details of the network architecture by running only the first bracket of Hyperband with $\eta = 3$ and $S = 3$.

**Evaluation:** the best configuration found by Hyperband was trained five times with different random training/validation splits and evaluated on the test set to produce the final results, except for STL-10 where we used the official ten cross-validation folds.

## 4.2 RESULTS

The best performance achieved by our method is shown in Table 1 and compared with SelfPU (Chen et al., 2020b), DAN (Liu et al., 2019) and PAN (Hu et al., 2021). *PUUPL* was always able to improve over the baseline nnPU loss, with larger gap for more difficult datasets such as CIFAR-10 (+0.98%) and IMDb (+1.89%) as well as over SelfPU (Chen et al., 2020b) and DAN (Liu et al., 2019), setting a new state-of-the-art for PU learning. Moreover, *PUUPL* is naturally very well calibrated despite the absence of explicit calibration on labeled data (Fig. 2), making its predictions inherently reliable. The best pseudo-labeling hyperparameters constitute the defaults we suggested in Algorithm 1 and are $K = 2$, $T = 1000$, $t_l = 0.05$ and $t_u = 0.35$. Note that the baseline nnPU scores reported in Table 1 were also obtained by training an ensemble of two networks with the nnPU loss, thus possibly explaining the discrepancy observed with SelfPU. The best network architecture for IMDb is shown in Table A.2.

## 4.3 ABLATION STUDIES

We performed ablation studies on the CIFAR-10 dataset by changing one parameter at a time of the best configuration found by Hyperband, training and evaluating with five different splits and reporting the test accuracy corresponding to the best validation score for each run. To limit the computational resources needed, we used at most 15 pseudo-labeling iterations.

**Weights initialization:** We confirmed the observation that it is beneficial to re-initialize the weights after each pseudo-labeling step (Arazo et al., 2020), with slightly better performance (+0.052%) achieved when the weights are re-initialized to the same values before every pseudo-labeling iteration (Fig. 3a). We believe this encourages the model to be consistent across pseudo-labeling rounds.

**Pseudo-label assignment:** Soft pseudo-labels were preferred over hard ones (+0.75%). We found that our model was very well calibrated with ECEs as low as 0.05 on the labeled validation data (Fig. 2), indicating that the soft pseudo-labels they estimated were reliable training targets and that *post hoc* calibration was not necessary. Contrary to expectation, however, re-assigning all pseudo-labels at every iteration harmed performance (−0.12%); instead, pseudo-labels should be kept fixed after being assigned for the first time. A possible explanation is that fixed pseudo-labels prevent the

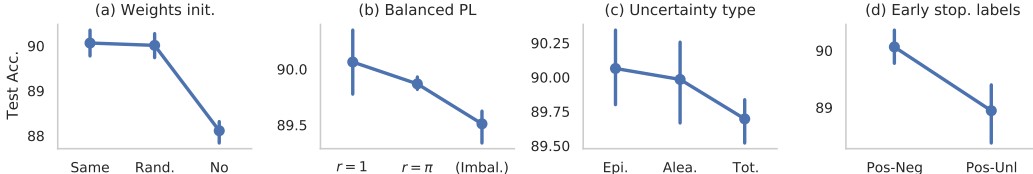

Figure 3: Mean and standard deviation of the test accuracy obtained over five runs by different variations of our *PUUPL* algorithm: (a) different weight initialization at each iteration, (b) balanced or imbalanced PL selection, (c) type of uncertainty, (d) whether to use positive-negative or positive-unlabeled validation set. Note the different scales on the $y$-axes.

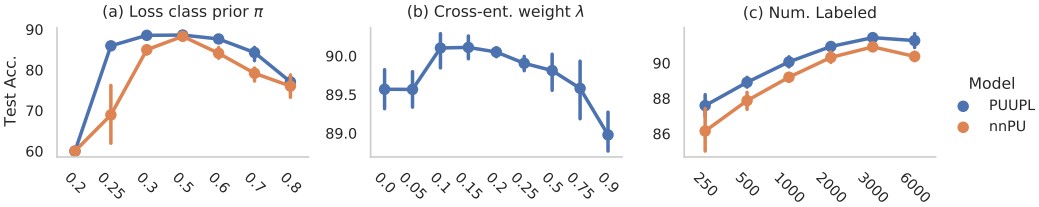

Figure 4: Mean and standard deviation of the test accuracy obtained over five runs by different hyperparameter values. Refer to the main text for a discussion. Note the different scales on the $y$-axes.

model's predictions from drifting too far away from the initial pseudo-labeling towards an incorrect assignment. It was also beneficial to assign the same number of positive and negative pseudo-labels (Fig. 3b) compared to keeping the same ratio $\pi$ of positives and negatives found in the whole dataset ($-0.20\%$) or not balancing the selection at all ($-0.55\%$).

**Uncertainty:** Ranking predictions by aleatoric performance was almost as good as ranking by epistemic uncertainty ($-0.08\%$), while total uncertainty produced moderately worse rankings ($-0.37\%$, Fig. 3c). An ensemble with only two networks achieved the best performance, while larger ensembles performed worse, and Monte Carlo dropout ($-0.85\%$) was better than ensembles of five ($-1.00\%$) and ten networks ($-1.58\%$).

**Early stopping:** Finally, performing early stopping on the validation PU loss resulted in worse accuracy ($-1.12\%$) compared to using the accuracy on positive-negative labels (Fig. 3d). Although considerable when compared to the impact of other algorithmic choices, such performance drop indicates that *PUUPL* can be used effectively in real-world scenarios with no labeled validation data available.

### 4.4 ROBUSTNESS

Following the same protocol as the ablation studies in Section 4.3, we tested the robustness of our method with respect to misspecifications of the continuous hyperparameters (Fig. 4).

**Pseudo-labeling:** Our method was fairly robust to the maximum number $T$ of assigned pseudo-labels and the maximum uncertainty threshold $t_l$ for the pseudo-labels, with almost constant performance up to $T = 1000$ and $t_l = 0.1$. The best performance was achieved by the combination having $T = 1000$ and $t_l = 0.05$, but both of these experiments were performed while disabling the other constraint (i.e., setting $T = \inf$ when testing $t_l$ and vice-versa). Using only a constraint on $T$ resulted in a reduction of $-0.11\%$, while constraining $t_l$ alone resulted in a reduction of $-1.04\%$. The results for $t_u$ were less conclusive as for the general trend, possibly because values lower than 0.35 require more than the 15 pseudo-labeling iterations we used for the experiment, and values above 0.4 did not show significant differences.

**Misspecification of the class prior:** The performance of our framework slowly degraded as the prior $\pi$ moved further from the true value of 0.4 with a performance reduction of less than 2.5% in the range $[0.3, 0.6]$ (Fig. 4a). Furthermore, the performance gap between *PUUPL* and nnPU widens as $\pi$ is more severely misspecified. Modern losses for PU learning such as uPU and nnPU rely on the

|  | nnPU | nnPUSB |
|---|---|---|
| Only PU loss | 87.05 (0.71) | 87.31 (0.60) |
| PU loss+*PUUPL* | **87.70 (0.69)** | **87.91 (0.71)** |

Table 2: Test accuracy of our framework on the CIFAR-10 dataset with a selection bias on the positive labels when using the nnPU and nnPUSB losses. Our framework improves over the base PU loss in both cases; in particular, *PUUPL* with nnPU loss achieved better performance than the nnPUSB loss alone.

correct estimation of the positive class prior $\pi$ from domain knowledge or *a priori* estimation of $\pi$, which constitutes a whole research branch in PU learning (Chen et al., 2020a) and is a significant challenge in any practical PU application (Bekker and Davis, 2020; Chen et al., 2020a). We believe that the inclusion of epistemic uncertainty, the usage of soft labels and the convex combination of two losses enables *PUUPL* to be considerably more robust to significant misspecification of the class prior $\pi$.

**Loss combination:** The best performing combination had $\lambda = 0.15$, with modest performance reduction until $\lambda = 0.5$ ($-0.25\%$, Fig. 4b). Values of 0.05 and below resulted in the same performance reduction of -0.5%, similarly to $\lambda = 0.75$, and performance was 1.09% worst at $\lambda = 0.9$. Too large $\lambda$ might facilitate the emergence of a harmful confirmation bias, but it is nonetheless important to train on the pseudo-labels, too, to avoid losing the information contained therein.

**Number of training labeled positives:** The performance of our method steadily increased and seemed to plateau at 91.4% between 3,000 and 6,000 labeled positives. The gap between nnPU and *PUUPL* is largest in the low labeled data region with a 1.44% gap at 250 labels, where we achieved 87.59% accuracy, shrinking to a gap of 0.52% with 3,000 labels, where our performance was 91.44% (Fig. 4c). This supports our intuition about the importance of uncertainty because, as the amount of labeled data decreases, uncertainty becomes more important to detect overfitting and to prevent the model from assigning incorrect pseudo-labels.

**Positive bias:** The most general assumption of PU learning is that the labeled examples are a biased sample from the positive distribution (Bekker and Davis, 2020). We tested *PUUPL* in such a biased setting where positives in the training and validation sets were with 50% chance an airplane, 30% chance an automobile, 15% chance ship and 5% chance truck, while in previous experiments the positives were evenly composed of airplanes, automobiles, ships and trucks. The test distribution was unchanged, meaning that test samples are half as likely to be airplanes compared to the training set, and five times more likely to be to be truck images. We also fixed all hyperparameters to the values identified previously, except for the loss $\mathcal{L}_{PU}$ where we used the nnPUSB loss (Kato et al., 2019) to handle the positive bias. The baseline with nnPUSB loss performed better than the nnPU loss (+0.26%), but worse than *PUUPL* with the nnPU loss (-0.39%), highlighting the benefit of our uncertainty-aware approach. The best performance was however achieved with *PUUPL* on top of the nnPUSB loss (+0.21% compared to nnPU and -2.27% compared to the unbiased setting), showing that *PUUPL* can leverage the advantages of different PU losses and further improve on them (Table 2).

## 5 CONCLUSIONS

In this paper, we proposed an uncertainty-aware pseudo-labeling framework for PU learning which quantifies the epistemic uncertainty of an ensemble of networks and selects which examples to pseudo-label based on their predictive uncertainty. We demonstrated the benefits of our approach on different data modalities and biased settings, achieving state-of-the-art performance in all our benchmarks. We further conducted extensive ablation studies and investigated the robustness of our approach, showing it to be reliable in settings that are likely to be encountered in the real world, such as a bias in the positive data, the unavailability of labeled negatives as validation data and the misspecification of the class prior $\pi$.

**Ethics statement:** Most of the ethical concerns stem from the specific application and dataset. Here we have shown a certain robustness towards biased positive labels without providing a comprehensive assessment, therefore practitioners should always ensure, insofar as possible, that the obtained

predictions are "fair" (with "fairness" defined appropriately w.r.t. the target application) and do not systematically affect particular subsets of the population of interest.

**Reproducibility statement:** The source code for the framework is available in the supplementary material.

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

| Hyper-parameter | Value range |
|---|---|
| Estimator | Ensemble or MC Dropout |
| Number of samples | $[2, 25]$ |
| Uncertainty type | Aleatoric, epistemic, total |
| Max. new labels $T$ | $[100, 5000]$ |
| Max. new label uncertainty $t_l$ | $[0, -\log 2]$ |
| Min. unlabel uncertainty $t_u$ | $[0, -\log 2]$ |
| Reassign all pseudo-labels | Yes or no |
| Re-initialize to same weights | Yes or no |
| Cross-entropy weight $\lambda$ | $[0, 1]$ |

Table A.1: Pseudo-labeling hyperparameters

| Layer type | Layer parameters |
|---|---|
| LSTM | inpus size=200, hidden size=128, num layers=2, dropout=0.25, bidirectional=True |
| Dropout | p=0.2 |
| Linear | in features=256, out features=196, bias=True |
| Batch Norm. | eps=1e-05, momentum=0.1 |
| ReLU | |
| Dropout | p=0.2 |
| Linear | in features=196, out features=196, bias=True |
| Batch Norm. | eps=1e-05, momentum=0.1 |
| ReLU | |
| Linear | in features=196, out features=1, bias=True |

Table A.2: Network architecture used for the IMDb experiments

## APPENDIX

## A    NETWORK ARCHITECTURE AND HYPERPARAMETERS

Table A.1 reports the hyperparameters related to pseudo-labeling and their ranges. Table A.2 reports the network architecture used in the IMDb experiments, while Table A.3 reports the network used with CIFAR-10.

## B    DATASET INFORMATION

Table B.1 reports the number of samples for each split and each dataset. For the image datasets, we subtracted the mean pixel intensity in the training set and divided by the standard deviation. For IMDb we used pre-trained GloVe embeddings of size 200 on a corpus of six billion tokens.

| Layer type | Layer parameters |
|---|---|
| Conv. 2D | in channels=3, out channels=96, kernel size=3 stride=1, padding=1 |
| Dropout | p=0.15 |
| Batch Norm. | eps=1e-05, momentum=0.1 |
| ReLU | |
| Conv. 2D | in channels=96, out channels=96, kernel size=3, stride=1, padding=1 |
| Dropout | p=0.15 |
| Batch Norm. | eps=1e-05, momentum=0.1 |
| ReLU | |
| Conv. 2D | in channels=96, out channels=96, kernel size=3, stride=2, padding=1 |
| Dropout | p=0.15 |
| Batch Norm. | eps=1e-05, momentum=0.1 |
| ReLU | |
| Conv. 2D | in channels=96, out channels=192, kernel size=3, stride=1, padding=1 |
| Dropout | p=0.15 |
| Batch Norm. | eps=1e-05, momentum=0.1 |
| ReLU | |
| Conv. 2D | in channels=192, out channels=192, kernel size=3, stride=1, padding=1 |
| Dropout | p=0.15 |
| Batch Norm. | eps=1e-05, momentum=0.1 |
| ReLU | |
| Conv. 2D | in channels=192, out channels=192, kernel size=3, stride=2, padding=1 |
| Dropout | p=0.15 |
| Batch Norm. | eps=1e-05, momentum=0.1 |
| ReLU | |
| Conv. 2D | in channels=192, out channels=192, kernel size=3, stride=1, padding=1 |
| Dropout | p=0.15 |
| Batch Norm. | eps=1e-05, momentum=0.1 |
| ReLU | |
| Conv. 2D | in channels=192, out channels=192, kernel size=1, stride=1, padding=0 |
| Dropout | p=0.15 |
| Batch Norm. | eps=1e-05, momentum=0.1 |
| ReLU | |
| Conv. 2D | in channels=192, out channels=10, kernel size=1, stride=1, padding=0 |
| Dropout | p=0.15 |
| Batch Norm. | eps=1e-05, momentum=0.1 |
| ReLU | |
| Flatten | |
| Linear | in features=640, out features=1000, bias=True |
| ReLU | |
| Linear | in features=1000, out features=1000, bias=True |
| ReLU | |
| Linear | in features=1000, out features=1, bias=True |

Table A.3: Network architecture used for the CIFAR-10 experiments

| Dataset | Tot. Pos. | Tot. Neg. | Train Lab. | Train Unlab. | Val. size | Test Size |
|---|---|---|---|---|---|---|
| MNIST | 30,508 | 29,492 | 1,000 | 54,000 | 5,000 | 10,000 |
| F-MNIST | 30,000 | 30,000 | 1,000 | 54,000 | 5,000 | 10,000 |
| CIFAR-10 | 20,000 | 30,000 | 1,000 | 44,000 | 5,000 | 10,000 |
| STL-10 | ? | ? | 3,600 | 105,400 | 5,000 | 8,000 |
| IMDb | 12,500 | 12,500 | 1,000 | 19,000 | 5,000 | 25,000 |

Table B.1: Total size of the datasets and the data splits. As for STL-10, we used the entire set of examples, incuding the truly unlabeled ones.

