# OpenReview forum: "Positive-Unlabeled Learning with Uncertainty-aware Pseudo-label Selection"
_ICLR.cc/2022/Conference — ICLR 2022 Submitted_

### Official Review · Reviewer_kfAD · 2021-10-30

**Correctness:** 2
**Technical Novelty And Significance:** 2
**Empirical Novelty And Significance:** 2
**Recommendation:** 3
**Confidence:** 5

**Main Review:**

This paper studies the PU learning problem. It proposes a two-step approach that can estimate the pseudo-label uncertainty so that more reliable pseudo-labels can be assigned, which improves the predictive performance. The proposed estimation method is different from previous methods.

My main concern is that two key assumptions are made in the paper, which are not realistic. The first one is that a positive and negative labeled validation set is available and used as the validation set for early stopping. The second is that the proportion of positive and negative samples is known. Both are problematic. The paper justifies the assumptions by listing some prior work that made one or the other assumption. However, I don’t think the fact that some prior work using these assumptions makes the assumptions right. In a real-life situation, these numbers are not available. There are many approaches that do not make any of the assumptions, but they are not compared, e.g., the recent system PAN, “Predictive adversarial learning from positive and unlabeled data,” AAAI-2021.

Prior work has used several techniques to score the unlabeled set and then label them. Since the paper that proposed the two-step approach, “partially supervised classification of text documents,” ICML-2002, several approaches have been proposed to pseudo-label the unlabeled set. I understand that the proposed approach is different and probably more advanced than those older approaches, but it will be more complete to have an experimental comparison with some earlier approaches to show the advantage of the proposed method.

The paper is easy to understand.


**Summary Of The Paper:**

This paper studies the PU learning problem. It proposes a two-step approach that can estimate the pseudo-label uncertainty so that more reliable pseudo-labels can be assigned, which improves the predictive performance. The proposed estimation method is different from previous methods.

**Summary Of The Review:**

Although the proposed estimation method is different from existing ones, my main concern is that the two assumptions are unrealistic. They actually run against the very setting of PU learning. Without the two assumptions, the proposed method would have problems. The paper should also compare with some earlier methods.

---

> ### Author Response · Authors · 2021-11-17
> **Response to Reviewer kfAD [1/2]**
>
> Thank you for your comments.
>
> > a positive and negative labeled validation set is available and used as the validation set for early stopping
>
> We would like to point out that a cleanly labeled validation set does not constitute a hard requirement for our method. We specifically elaborated on that at the end of section 4.3 where we showed that PUUPL’s performance only decreases negligibly when early stopping is performed using the validation nnPU loss which requires *no* labeled negatives. Hence, this is not a deal-breaker in practice.
>
> However, assuming access to a labeled validation set *in a scientific evaluation* can be justified in several ways:
> 1. Using a labeled validation set is necessary to provide a fair comparison with other methods that use it. See for instance the paragraph after equation 6 in section 3.2 of Self-PU by Chen (ICML 2020)  [1]: “[...] on the validation set which contains clean positive and negative examples, [...]”. Other examples for the use of a cleanly labeled validation set for methodological research in PU learning come from Tanaka (2018, CVPR) [2]: “ We determined the [...] hyper parameters [...] based on the validation accuracy.” (Section 5.2) and the paper PAN by Hu et al. (AAAI 2021) [3] suggested by reviewer kfAD which reports the best test accuracy over 200 epochs (section 5.2), which in essence corresponds to early stopping performed on the test set (we use separate sets for stopping and evaluating).
> 2. A labeled validation set is required for a trustworthy evaluation of the *potential* of the method and accepted as such in the field of semi-supervised learning [4]. Many other works simply report hyperparameters without mentioning how they were found, or justify them through poorly explained ablation studies that still require labeled data.  We instead followed a rigorous and clearly documented experimental protocol, optimizing hyperparameters on the validation set and performing the final evaluation on a separate test set, thus ensuring that our benchmarks were not tainted by any sort of information leakage which would result in artificially over-optimistic figures. Even with such a strict protocol we were able to achieve SotA performance. We feel that an explicit description of the use of such a validation set is a good practice in order to nurture further research in this field and make sure methods are evaluated fairly and consistently.
>
> To make it clear that our method in and of its own does not necessitate a labeled validation set, but such a set of examples is necessary to conduct a proper evaluation, we moved the “early stopping” section to the experimental setup.
>
> > the proportion of positive and negative samples is known
>
> We agree that this is a limitation of our method as presented in the paper. However, there are methods that estimate the class prior exclusively from positive-unlabeled data [5] and class prior estimation in itself is a separate research branch in PU Learning [6]. We provided an experiment in section 4.4 showing that the performance penalty for misspecification of the class prior is relatively minor.
>
> Most importantly, however, our framework in and of its own does not necessitate a known class prior. Its necessity stems from our use of the nnPU and nnPUSB losses in our experiments, but our approach should instead be seen as a general blueprint for developing positive-unlabeled classifiers based on the technique of pseudo-labeling. In fact, as we highlighted in the contributions, PUUPL can use any loss function developed for PU learning, including those that do not need this prior to be specified, such as VPU [7]. Further future advancement would also be trivial to integrate.
>
> > In a real-life situation, these numbers are not available.
>
> From our perspective, the real world is highly varied, multi-faceted and unpredictable, and it is conceivable that in certain situations either one of our assumptions would be validated by circumstances that are unforeseeable *a priori*. In such cases, practitioners may prefer to forego alternative complicated training procedures [8] or rather employ methods such as ours that are simpler to understand, tune, and deploy. It is generally advantageous to make use of appropriate inductive biases when appropriate and justified by the specific practical application, thus we feel there is room for methods that are simpler, but less broadly applicable.

---

> > ### Author Response · Authors · 2021-11-17
> > **Response to Reviewer kfAD [2/2]**
> >
> >
> > > There are many approaches that do not make any of the assumptions, but they are not compared, e.g., the recent system PAN, “Predictive adversarial learning from positive and unlabeled data,” AAAI-2021.
> >
> > Thank you for referring us to PAN [3], we now report their performance in our experimental comparison, to which we added the performance of an uncertainty-unaware pseudo-labeling baseline. Preliminary results are as follow:
> >
> > - MNIST - nnPU: 95.26 - PL: 95.78 - PUUPL 96.01
> > - F-MNIST - nnPU: 95.70 - PL: 95.84 - PUUPL: 95.91
> > - CIFAR-10 - nnPU: 89.20 - PAN: 89.70 - PL: 89.74 - PUUPL: 90.18
> > - IMDb - nnPU:78.57 - PAN: 78.84 (1,250 labeled positives instead of our 1,000) - PL: 80.64 - PUUPL: 81.83
> >
> > However, we would like to draw attention to the fact that PAN still uses early stopping: “we give [...] the best accuracy for each system on each dataset over 200 epochs [...] over 5 runs” (section 5.2). We feel that our (perhaps questionable) assumption of a labeled validation set, used to conduct a proper experimental evaluation and not as a hard methodological requirement, is scientifically more sound compared to optimizing a model directly on the test set (which also contains clean negative samples).
> >
> >
> > [1] Chen, X., Chen, W., Chen, T., Yuan, Y., Gong, C., Chen, K., & Wang, Z. (2020, November). Self-pu: Self boosted and calibrated positive-unlabeled training. In International Conference on Machine Learning (pp. 1510-1519). PMLR.
> >
> > [2] Tanaka, Daiki, Daiki Ikami, T. Yamasaki and Kiyoharu Aizawa. “Joint Optimization Framework for Learning with Noisy Labels.” 2018 IEEE/CVF Conference on Computer Vision and Pattern Recognition (2018): 5552-5560.
> >
> > [3] Hu, W., Le, R., Liu, B., Ji, F., Ma, J., Zhao, D., & Yan, R. (2021, May). Predictive Adversarial Learning from Positive and Unlabeled Data. In Proceedings of the AAAI Conference on Artificial Intelligence (Vol. 35, No. 9, pp. 7806-7814).
> >
> > [4] Oliver, Avital, Augustus Odena, Colin Raffel, Ekin Dogus Cubuk and Ian J. Goodfellow. “Realistic Evaluation of Deep Semi-Supervised Learning Algorithms.” NeurIPS (2018).
> >
> > [5] Marthinus C. du Plessis, Gang Niu, and Masashi Sugiyama. Class-prior estimation for learning from positive and unlabeled data. 106(4):463–492, nov 2016
> >
> > [6] Bekker, Jessa and Jesse Davis. “Learning from positive and unlabeled data: a survey.” Machine Learning 109 (2020): 719-760.
> >
> > [7] Chen, Hui, Fangqing Liu, Yin Wang, Liyue Zhao and Hao Wu. “A Variational Approach for Learning from Positive and Unlabeled Data.” NeurIPS 2020
> >
> > [8] Sculley, D., Gary Holt, Daniel Golovin, Eugene Davydov, Todd Phillips, Dietmar Ebner, Vinay Chaudhary, Michael Young, Jean-François Crespo and Dan Dennison. “Hidden Technical Debt in Machine Learning Systems.” NIPS (2015).

---

### Official Review · Reviewer_3wif · 2021-11-01

**Correctness:** 3
**Technical Novelty And Significance:** 2
**Empirical Novelty And Significance:** 2
**Recommendation:** 3
**Confidence:** 4

**Main Review:**

The idea of using the epistemic uncertainty in PU learning is interesting. However, it is not clear the technical novelty and it lacks a simple baseline against uncertainty-aware pseudo-labeling.

---

What is the difficulty to introduce uncertainty-aware pseudo-labeling techniques to PU learning? It seems that simply applying uncertainty-aware pseudo-labeling technique results in the proposed method. That is, the technical novelty of the proposed method seems not strong. If the authors can clarify non-trivial technical contributions, it would help understand this paper well.

Comparison with uncertainty-unaware pseudo-labeling techniques helps understand the superiority of the proposed method. It is expected to illustrate that considering uncertainty is important when we use pseudo-labeling in PU learning. For example, nnPU with ordinary pseudo-labeling techniques could be compared with nnPU with the proposed uncertainty-aware pseudo-labeling.

In Section 3.2, there is the sentence "we assume cleanly labeled (positives and negatives) validation and test sets (see Section 3.6) as usual in literature." I checked the cited papers quickly, but could not find such a setting. If negative samples are available, we can use them (e.g., a half of them) in training. And, it might result in better performance thanks to negative samples. So, the assumption sounds not practical. Thus, the results reported in the experiments are not good evidence of the effectiveness of the proposed method. But, there is a chance that I miss the sentences supporting the authors' assumption. If the authors can show the sentences describing the assumption about the validation set, I will confirm it.

=== After rebuttal ===
Thank you for your responses.
The idea of using the epistemic uncertainty in PU learning is interesting. So, I hope that the authors will resolve my and the others' concerns in the next venue.

**Summary Of The Paper:**

This paper proposes PUUPL, an uncertainty-aware pseudo-label selection method for positive-unlabeled (PU) learning. To improve the performance of pseudo-labeling, the authors suggest using the epistemic uncertainty (the difference between the entropy of the mean prediction and the mean of entropies of each prediction). In the experiments, PUUPL outperformed the existing state-of-the-art PU learning methods.

**Summary Of The Review:**

This paper introduces uncertainty-aware pseudo-labeling techniques to PU learning. Although the idea is interesting, the technical novelty and the performance improvement seem marginal. In addition, according to the paper, it assumes cleanly-labeled validation data (we can access negative samples) are available, which is a strange assumption in PU learning.

---

> ### Author Response · Authors · 2021-11-16
> **Response to Reviewer 3wif [1/2]**
>
> Thank you for your comments.
>
> > it lacks a simple baseline against uncertainty-aware pseudo-labeling. [...] nnPU with ordinary pseudo-labeling techniques could be compared with nnPU with the proposed uncertainty-aware pseudo-labeling.
>
> We are training such a baseline and will soon update the manuscript. Preliminary results are as follows (PL refers to the uncertainty-unaware baseline):
>
> - MNIST - nnPU: 95.26 - PL: 95.78 - PUUPL 96.01
> - F-MNIST - nnPU: 95.70 - PL: 95.84 - PUUPL: 95.91
> - CIFAR-10 - nnPU: 89.20 - PAN: 89.70 - PL: 89.74 - PUUPL: 90.18
> - IMDb - nnPU:78.57 - PAN: 78.84 (1,250 labeled positives instead of our 1,000) - PL: 80.64 - PUUPL: 81.83
>
> > the technical novelty of the proposed method seems not strong
>
> We would like to draw attention to four arguments that in our opinion make PUUPL a worthy contribution to the field:
> 1. To the best of our knowledge, PUUPL is the *first* uncertainty-aware approach in positive-unlabeled learning. Quantification of predictive uncertainty is especially important in *safety-critical domains* such as clinical applications where positive-unlabeled data is common and traditional supervised methods are not applicable.
> 2. We present a *simpler*, *more robust* and *rigorously evaluated* method that *outperforms* previous approaches for PU learning and does not require complex training strategies. In our opinion, simplicity is an important dimension to consider [1] and such an improvement over the current SOTA is indeed novel.
> 3. We provide a comprehensive and clearly documented *experimental procedure* to quantify the impact of each algorithmic decision on the final performance via *extensive* ablation and robustness studies that span more than 12,000 GPU-hours.
> 4. The result of our experimental studies are robust hyperparameters that perform well across datasets, data modalities and neural architectures, thus sparing end users from the burden of tuning our solution.
>
> > it assumes cleanly-labeled validation data (we can access negative samples) are available, which is a strange assumption in PU learning. [...]
>
> We would like to point out that a cleanly labeled validation set does not constitute a hard requirement for our method. We specifically elaborated on that at the end of section 4.3 where we showed that PUUPL’s performance only decreases negligibly when early stopping is performed using the validation nnPU loss which requires *no* labeled negatives. Hence, this is not a deal-breaker in practice.
>
> However, assuming access to a labeled validation set *in a scientific evaluation* can be justified in several ways:
> 1. Using a labeled validation set is necessary to provide a fair comparison with other methods that use it. See for instance the paragraph after equation 6 in section 3.2 of Self-PU by Chen (ICML 2020)  [2]: “[...] on the validation set which contains clean positive and negative examples, [...]”. Other examples for the use of a cleanly labeled validation set for methodological research in PU learning come from Tanaka (2018, CVPR) [3]: “ We determined the [...] hyper parameters [...] based on the validation accuracy.” (Section 5.2) and the paper PAN by Hu et al. (AAAI 2021) [4] suggested by reviewer kfAD which reports the best test accuracy over 200 epochs (section 5.2), which in essence corresponds to early stopping performed on the test set (we use separate sets for stopping and evaluating).
> 2. A labeled validation set is required for a trustworthy evaluation of the *potential* of the method and accepted as such in the field of semi-supervised learning [5]. Many other works simply report hyperparameters without mentioning how they were found, or justify them through poorly explained ablation studies that still require labeled data.  We instead followed a rigorous and clearly documented experimental protocol, optimizing hyperparameters on the validation set and performing the final evaluation on a separate test set, thus ensuring that our benchmarks were not tainted by any sort of information leakage which would result in artificially over-optimistic figures. Even with such a strict protocol we were able to achieve SotA performance. We feel that an explicit description of the use of such a validation set is a good practice in order to nurture further research in this field and make sure methods are evaluated fairly and consistently.
>
> To make it clear that our method in and of its own does not necessitate a labeled validation set, but such a set of examples is necessary to conduct a proper evaluation, we moved the “early stopping” section to the experimental setup.

---

> > ### Author Response · Authors · 2021-11-16
> > **Response to Reviewer 3wif [2/2]**
> >
> > > If the authors can show the sentences describing the assumption about the validation set, I will confirm it.
> >
> > 1. Self-PU by Chen (ICML 2020)  [2]: “[...] on the validation set which contains clean positive and negative examples, [...]”.
> > 2. Tanaka (CVPR 2018) [3]: “ We determined the [...] hyper parameters [...] based on the validation accuracy.” (Section 5.2)
> > 3. PAN by Hu et al. (AAAI 2021) [4] suggested by reviewer kfAD which reports the best test accuracy over 200 epochs (section 5.2), which corresponds to using the cleanly labeled test set as validation *and* test set at the same time.
> >
> > > the performance improvement seems marginal.
> >
> > We would like to draw attention to the fact that other recent PU methods show improvements of similar magnitude on the same standard benchmark datasets. For instance, Self-PU [2] was reported to outperform training with the nnPU Loss by 1.08% percentage points and PAN [4] by 1.10% percentage points on CIFAR-10 with 1,000 positive samples. The improvement shown by PUUPL in the same setting amounts to 1.58% percentage points improvement over the nnPU baseline.
> >
> > Furthermore, we urge reviewers to consider that we were able to substantially improve over the state-of-the-art in spite of the simplicity of our method. In our opinion this should be considered as a strength rather than a weakness not only because of its philosophical underpinning in the Occam’s razor principle but also because of the well-documented issues arising from using overly complicated machine learning systems in the real-world [1].
> >
> > [1] Sculley, D., Gary Holt, Daniel Golovin, Eugene Davydov, Todd Phillips, Dietmar Ebner, Vinay Chaudhary, Michael Young, Jean-François Crespo and Dan Dennison. “Hidden Technical Debt in Machine Learning Systems.” NIPS (2015).
> >
> > [2] Chen, X., Chen, W., Chen, T., Yuan, Y., Gong, C., Chen, K., & Wang, Z. (2020, November). Self-pu: Self boosted and calibrated positive-unlabeled training. In International Conference on Machine Learning (pp. 1510-1519). PMLR.
> >
> > [3] Tanaka, Daiki, Daiki Ikami, T. Yamasaki and Kiyoharu Aizawa. “Joint Optimization Framework for Learning with Noisy Labels.” 2018 IEEE/CVF Conference on Computer Vision and Pattern Recognition (2018): 5552-5560.
> >
> > [4] Hu, W., Le, R., Liu, B., Ji, F., Ma, J., Zhao, D., & Yan, R. (2021, May). Predictive Adversarial Learning from Positive and Unlabeled Data. In Proceedings of the AAAI Conference on Artificial Intelligence (Vol. 35, No. 9, pp. 7806-7814).
> >
> > [5] Oliver, Avital, Augustus Odena, Colin Raffel, Ekin Dogus Cubuk and Ian J. Goodfellow. “Realistic Evaluation of Deep Semi-Supervised Learning Algorithms.” NeurIPS (2018).

---

> > > ### Comment · Reviewer_3wif · 2021-11-19
> > > **Comments to responses**
> > >
> > > Thank you for your responses.
> > >
> > > Tanaka et al. (CVPR2018) is work for noisy-label learning. So, it seems a different setting. It would not be appropriate to cite this paper to support the validity of using a cleanly labeled dataset in PU learning.
> > >
> > > I could not find the exact sentence describing the cleanly labeled dataset in Hu et al. (AAI2021). In Section 5.2, it says that the performance at 200 epochs was reported, but it is not saying that they stopped training by using a cleanly labeled dataset. Could you please cite the sentence that mentions a cleanly labeled dataset?
> > >
> > > Self-PU paper says that the validation set contains a cleanly labeled dataset. However, if we have a cleanly labeled dataset, it would be recommended to use them for training in real-world applications. We can use half of them for training and the rest for validation. Also, since the authors claim that the proposed method does not require a cleanly labeled dataset, it is better to report all the results in experiments without using a cleanly labeled dataset. If the performance of Self-PU is a bit better than the other methods, it would be the power of using a cleanly labeled dataset in the validation.
> > >
> > > I asked what is the difficulty to introduce the uncertainty-aware pseudo-labeling technique to PU learning. This paper seems the first paper to introduce the uncertainty-ware pseudo-labeling technique to PU learning. However, pseudo-labeling techniques have been applied to several tasks. So, I was wondering how difficult introducing the new technique is and what is the special idea from the authors. If possible, for each section, it is helpful to show whether the ideas come from existing studies or the authors. Then, I can easily understand what the authors' technical contributions are.

---

> > > > ### Author Response · Authors · 2021-11-29
> > > > **Further responses**
> > > >
> > > > > I could not find the exact sentence describing the cleanly labeled dataset in Hu et al. (AAI2021). In Section 5.2, it says that the performance at 200 epochs was reported, but it is not saying that they stopped training by using a cleanly labeled dataset. Could you please cite the sentence that mentions a cleanly labeled dataset?
> > > >
> > > > The sentence is: "Since different epochs give different results, for a fair comparison, we we give the average of both the best F-score (F) and best accuracy (Acc) for each system on each dataset over 200 epochs (all systems converged before 200 epochs) over 5 runs". This means that they reported the best test accuracy obtained in the course 200 epochs, averaged over five repetitions. This is early stopping, on the labeled test set.
> > > >
> > > > > However, if we have a cleanly labeled dataset, it would be recommended to use them for training in real-world applications. We can use half of them for training and the rest for validation.
> > > >
> > > > This is easily done with our method: the L set should be initialized with the available labeled samples, rather than empty.
> > > >
> > > > > Also, since the authors claim that the proposed method does not require a cleanly labeled dataset, it is better to report all the results in experiments without using a cleanly labeled dataset.
> > > >
> > > > We reported such results at the end of section 4.3, in the paragraph titled "early stopping".
> > > >
> > > > > Then, I can easily understand what the authors' technical contributions are.
> > > >
> > > > Our technical contribution is outperforming complicated alternatives with a simpler method.

---

> > > > > ### Comment · Reviewer_3wif · 2021-11-29
> > > > > **Reply to Further responses**
> > > > >
> > > > > > This is early stopping, on the labeled test set.
> > > > >
> > > > > Thank you for pointing it out. If so, it might be different from clean **validation** dataset.
> > > > >
> > > > > ---
> > > > >
> > > > > > We reported such results at the end of section 4.3, in the paragraph titled "early stopping".
> > > > >
> > > > > Thank you, but I wrote
> > > > > > ... it is better to report **all** the results in experiments ...
> > > > >
> > > > > ---
> > > > >
> > > > > > Our technical contribution is outperforming complicated alternatives with a simpler method.
> > > > >
> > > > > Thanks. I have already understood this point, but I would be grateful if I could know technical differences. The proposed method might be simple, but I guess that there are certain technical differences compared with the existing studies. I am asking that points and wrote the following reply:
> > > > > > I was wondering how difficult introducing the new technique is and what is the special idea from the authors. If possible, for each section, it is helpful to show whether the ideas come from existing studies or the authors. Then, I can easily understand what the authors' technical contributions are.

---

### Official Review · Reviewer_G93N · 2021-11-02

**Correctness:** 3
**Technical Novelty And Significance:** 2
**Empirical Novelty And Significance:** 2
**Recommendation:** 5
**Confidence:** 4

**Details Of Ethics Concerns:**

No ethics concerns.

**Main Review:**

Strengths
+ A simple uncertainty-aware pseudo-labeling framework for PU learning is proposed.
+ The proposed method is general and can be integrated into any loss function for PU learning.
+ This paper is easy to follow and the logic is clear.

Weaknesses
+ The novelty of this paper may not be enough. Specifically, the pseudo-labeling techniques have been well studied in a lot of weakly surprised learning scenarios such as label noise learning and semi-supervised learning.
+ The motivation of the metric used to quantify the uncertainty is not clear. The authors should give more explanations of the advantage of the proposed quantified metric.
+ The notations in Section 3.3 (which is the most important section) may cause confusion. For example, is it a sample x_i or example x_i? Is \hat{p}_{i} a vector or scalar? How about \hat{p}_{ik}?
+ Only four baseline methods are used in experiments. I think it is better to add more.

**Summary Of The Paper:**

The authors proposed to use pseudo-labels based on high confidence predictions to improve the classification performance of PU learning. Experiments show the effectiveness of the proposed method.

**Summary Of The Review:**

I think the research problem is interesting. However, I think the technical novelty is marginally significant, and the motivation of the proposed metric may not be clear.

---

> ### Author Response · Authors · 2021-11-16
> **Response to Reviewer G93N**
>
> Thank you for your comments.
>
> > The novelty of this paper may not be enough.
>
> We would like to draw attention to four arguments that in our opinion make PUUPL a worthy contribution to the field:
> 1. To the best of our knowledge, PUUPL is the *first* uncertainty-aware approach in positive-unlabeled learning. Quantification of predictive uncertainty is especially important in *safety-critical domains* such as clinical applications where positive-unlabeled data is common and traditional supervised methods are not applicable.
> 2. We present a *simpler*, *more robust* and *rigorously evaluated* method that *outperforms* previous approaches for PU learning and does not require complex training strategies. In our opinion, simplicity is an important dimension to consider [1] and such an improvement over the current SOTA is indeed novel.
> 3. We provide a comprehensive and clearly documented *experimental procedure* to quantify the impact of each algorithmic decision on the final performance via *extensive* ablation and robustness studies that span more than 12,000 GPU-hours.
> 4. The result of our experimental studies are robust hyperparameters that perform well across datasets, data modalities and neural architectures, thus sparing end users from the burden of tuning our solution.
>
> > The motivation of the metric used to quantify the uncertainty is not clear.
>
> We added a motivation for using such a metric in section 3.3, which now reads as follows: “Epistemic uncertainty corresponds to the mutual information between the parameters of the model and the true label of the sample. Low epistemic uncertainty thus means that that the model parameters would not change significantly if trained on the true label, suggesting that the prediction is indeed correct. Using such a prediction as target in the cross entropy loss would in turn provide a stronger, more explicit learning signal to the model, such that a correctly pseudo-labeled example provides a larger decrease in empirical risk compared to using the same example without any label within the positive-unlabeled loss.”
>
> > The notations in Section 3.3 (which is the most important section) may cause confusion.
>
> We clarified our notation in section 3.1.We used \hat{p}{ik} to denote the logit prediction of network k for example i, and \hat{p}{i} to denote the logits averaged across the K networks. Thus, both quantities are scalars.
>
> > Only four baseline methods are used in experiments. I think it is better to add more.
>
> As suggested by Reviewer 3wif, we are training an uncertainty-unaware pseudo-labeling baseline to show the benefit of introducing uncertainty in the procedure. We will also report the performance of PAN as suggested by reviewer kfAD. Preliminary results are as follows (PL refers to the uncertainty-unaware baseline):
>
> - MNIST - nnPU: 95.26 - PL: 95.78 - PUUPL 96.01
> - F-MNIST - nnPU: 95.70 - PL: 95.84 - PUUPL: 95.91
> - CIFAR-10 - nnPU: 89.20 - PAN: 89.70 - PL: 89.74 - PUUPL: 90.18
> - IMDb - nnPU:78.57 - PAN: 78.84 (1,250 labeled positives instead of our 1,000) - PL: 80.64 - PUUPL: 81.83
>
> [1] Sculley, D., Gary Holt, Daniel Golovin, Eugene Davydov, Todd Phillips, Dietmar Ebner, Vinay Chaudhary, Michael Young, Jean-François Crespo and Dan Dennison. “Hidden Technical Debt in Machine Learning Systems.” NIPS (2015).

---

### Official Review · Reviewer_9jfD · 2021-11-02

**Correctness:** 3
**Technical Novelty And Significance:** 1
**Empirical Novelty And Significance:** 2
**Recommendation:** 3
**Confidence:** 4

**Main Review:**

Trivial motivation:

In Abstract, the authors say that "two-steps procedures are vulnerable to incorrectly estimated pseudo-labels" and to mitigate this issue they propose this method. It is well known that many important PU learning methods are not two-step, such as nnPU.

Limited novelty:

First, I am not convinced that the proposed method is novel. The key techniques used are transferred from some existing methods (from Sec.3.2-3.6).
Second, I don't understand the meaning of designing an algorithm like this. For example, it uses an unbiased risk (Kiryo et al., 2017), which can already guarantee the consistency of learning. But complicating the learning procedure makes this loss lose its original advantage, and an additional regularization method has to be added. And it labels the instance according to the epistemic uncertainty (Eq.4-6). It is unclear why this measure is used and what the advantages are. The algorithm design is almost at random, at least from the current description. In addition, I think the complex algorithm design does not result in significant performance improvements.

Minor concerns:

- What is Eq.(3) for?
- The writing of the paper can be improved, e.g., "two-steps procedures"->"two-step procedures"


**Summary Of The Paper:**

This paper proposed a method to learn with PU data which quantifies the epistemic uncertainty of an ensemble of networks and selects which examples to pseudo-label based on their predictive uncertainty. The authors propose to use the pseudo-labeling technique based on the uncertainty of the prediction, combined with early stopping.

**Summary Of The Review:**

The novelty is limited and the contributions are incremental.

---

> ### Author Response · Authors · 2021-11-16
> **Response to Reviewer 9jfD**
>
> Thank you for your comments.
>
> > Trivial motivation
>
> We rephrased the motivation in the abstract to highlight the fact that the latest two-step techniques generally work better. It now reads: “Recent approaches addressed this problem via cost-sensitive learning by developing unbiased loss functions, and their performance was later improved by iterative pseudo-labeling solutions.”
>
> > It is unclear why this measure is used and what the advantages are.
>
> We added a motivation for using such a metric in section 3.3, which now reads as follows: “Epistemic uncertainty corresponds to the mutual information between the parameters of the model and the true label of the sample. Low epistemic uncertainty thus means that that the model parameters would not change significantly if trained on the true label, suggesting that the prediction is indeed correct. Using such a prediction as target in the cross entropy loss would in turn provide a stronger, more explicit learning signal to the model, such that a correctly pseudo-labeled example provides a larger decrease in empirical risk compared to using the same example without any label within the positive-unlabeled loss.”
>
> > But complicating the learning procedure makes this loss lose its original advantage, and an additional regularization method has to be added.
>
> The effectiveness of combining classification and positive-unlabeled losses as we do in Eq. 1 was demonstrated by Self-PU [1]. We do not introduce any new regularization methods beyond the standard dropout and weight decay; could you kindly clarify what regularization you are referring to?
>
> > Second, I don't understand the meaning of designing an algorithm like this. [...] The algorithm design is almost at random, at least from the current description.
>
> Our algorithm design follows two well-established principles:
> 1. Pseudo-labeling, which we improved upon by using uncertainty as the selection criterion. Related approaches are successfully applied in modern semi-supervised learning as suggested in [2,3,4] and in PU learning specifically by [1]. We explicitly stated this connection with pseudo-labeling throughout the manuscript, for example in the abstract: “[...] PUUPL, a new loss-agnostic training procedure for PU learning that incorporates epistemic uncertainty in pseudo-labeling.”, introduction: “[..] we propose a novel, uncertainty-aware pseudo-labeling framework for PU learning [...]”, methods: “We propose PUUPL [...], an iterative pseudo-labeling procedure to progressively select and label the most confident examples from unlabeled data” and conclusions: “[...] we proposed an uncertainty-aware pseudo-labeling framework for PU learning that [...]”
> 2. Extensive empirical analysis through ablation analysis, in order to provide further insights into our proposed  approach. In section 4.3 we provide empirical evidence that our proposed method achieves higher performance compared to reasonable alternative designs in terms of weights initialization, pseudo-label assignment and uncertainty measures.
>
> Hence, our design decisions are backed by empirical evidence and motivated by several successful methods and applications across semi-supervised learning and PU learning, as we described in the related work section. We thus disagree on the characterization of our method as “random design”, but there is a chance we misunderstood your comment. We would be happy to clarify our manuscript if you could concretely indicate the sections that do not sufficiently motivate our method.
>
> > I think the complex algorithm design does not result in significant performance improvements.
>
> We would like to draw attention to the fact that other recent PU methods show improvements of similar magnitude on the same standard benchmark datasets. For instance, Self-PU [1] was reported to outperform training with the nnPU Loss by 1.08% percentage points and PAN [5] by 1.10% percentage points on CIFAR-10 with 1,000 positive samples. The improvement shown by PUUPL in the same setting amounts to 1.58% percentage points improvement over the nnPU baseline.
>
> Furthermore, we urge reviewers to consider that we were able to substantially improve over the state-of-the-art in spite of the simplicity of our method. In our opinion this should be considered as a strength rather than a weakness not only because of its philosophical underpinning in the Occam’s razor principle but also because of the well-documented issues arising from using overly complicated machine learning systems in the real-world [6].
>
> > The writing of the paper can be improved.
>
> We corrected the typo and will conduct another proof-reading round.
>
> > What is Eq.(3) for?
>
> Eq. 3 is the sigmoid loss that we use in our experiments to compute the nnPU loss.

---

> > ### Author Response · Authors · 2021-11-16
> > **Literature for response to reviewer 9jfD**
> >
> > [1] Chen, X., Chen, W., Chen, T., Yuan, Y., Gong, C., Chen, K., & Wang, Z. (2020, November). Self-pu: Self boosted and calibrated positive-unlabeled training. In International Conference on Machine Learning (pp. 1510-1519). PMLR.
> >
> > [2] Rizve, M. N., Duarte, K., Rawat, Y. S., & Shah, M. (2021). In defense of pseudo-labeling: An uncertainty-aware pseudo-label selection framework for semi-supervised learning. In International Conference on Learning Representations.
> >
> > [3] Arazo, E., Ortego, D., Albert, P., O’Connor, N. E., & McGuinness, K. (2020, July). Pseudo-labeling and confirmation bias in deep semi-supervised learning. In 2020 International Joint Conference on Neural Networks (IJCNN) (pp. 1-8). IEEE.
> >
> > [4] Cascante-Bonilla, P., Tan, F., Qi, Y., & Ordonez, V. (2020). Curriculum labeling: Revisiting pseudo-labeling for semi-supervised learning. In Proceedings of the AAAI Conference on Artificial Intelligence.
> >
> > [5] Hu, W., Le, R., Liu, B., Ji, F., Ma, J., Zhao, D., & Yan, R. (2021, May). Predictive Adversarial Learning from Positive and Unlabeled Data. In Proceedings of the AAAI Conference on Artificial Intelligence (Vol. 35, No. 9, pp. 7806-7814).
> >
> > [6] Sculley, D., Gary Holt, Daniel Golovin, Eugene Davydov, Todd Phillips, Dietmar Ebner, Vinay Chaudhary, Michael Young, Jean-François Crespo and Dan Dennison. “Hidden Technical Debt in Machine Learning Systems.” NIPS (2015).

---

### Decision · Program_Chairs · 2022-01-20

**Decision:**

Reject

**Comment:**

This paper proposes a new loss-agnostic PU learning method based on uncertainty-aware pseudo-label selection.
I would like to thank the authors for their feedback to the initial reviews, which clarified many uncertain issues and improved our understanding of the current paper.
Nevertheless, even if the pseudo-labeling technique was applied to PU learning for the first time, given that it is a common practice in many weakly supervised learning tasks, the technical novelty is rather limited.

Therefore I cannot recommend acceptance of this paper.